# Active Travel of Czech and Polish Adolescents in Relation to Their Well-Being: Support for Physical Activity and Health

**DOI:** 10.3390/ijerph17062001

**Published:** 2020-03-18

**Authors:** Karel Frömel, Dorota Groffik, Josef Mitáš, Jan Dygrýn, Petr Valach, Michal Šafář

**Affiliations:** 1Faculty of Physical Culture, Palacký University Olomouc, 77111 Olomouc, Czech Republic; karel.fromel@upol.cz (K.F.); jan.dygryn@upol.cz (J.D.); michal.safar@upol.cz (M.Š.); 2Institute of Sport Science, The Jerzy Kukuczka Academy of Physical Education, 40-065 Katowice, Poland; d.groffik@awf.katowice.pl; 3Faculty of Education, University of West Bohemia, 301 00 Plzeň, Czech Republic; pvalach@ktv.zcu.cz

**Keywords:** IPAQ—long, recommendations, commuting, environment, secondary schools

## Abstract

The adoption of active travel (AT) habits in adolescence, supported by positive emotions, increases the chances of a lifelong positive attitude towards AT. The aim of this study was to assess the associations between active travel and well-being (WB), and to estimate the share of AT in weekly physical activity (PA) and its contribution to meeting the weekly PA recommendations in adolescents. The International Physical Activity Questionnaire—long form and the WHO-5 questionnaire were used to assess the level of AT, weekly PA and well-being of 2805 adolescents from 36 Czech and 39 Polish schools. A higher rate of AT is only significantly associated with higher well-being in girls. However, meeting AT recommendations in combination with higher WB increased the likelihood of meeting the weekly PA recommendations in both girls and boys. AT accounts for 22.5% of weekly PA of Czech (Polish 24.2%) boys. Concerning girls, it accounts for 24.9% of weekly PA in the Czech Republic and 24.5% in Poland. Meeting AT recommendations should be part of comprehensive school-based PA programs. State, school and municipal policies in the Central European region should pay more attention to the improvement of WB and the built environment for AT in secondary school adolescents.

## 1. Introduction

Active travel (AT) is an important part of adolescents’ everyday physical activity (PA). Nonetheless, we have been observing a long-term decreasing trend in AT throughout adolescence in many socioeconomically developed countries [1,2,3,4], but not in all of them [5]. The health, economic, social and behavioral benefits of AT are adequately supported by the research [6,7,8]. There is also strong evidence regarding the contribution of AT to the daily or weekly PA of adolescents, which is mostly based on subjective estimates of PA and, to a lesser extent, on the basis of objectively monitored PA [1]. The most frequent types of AT-related information are available with regard to walking and cycling [9,10,11]. Other popular types of AT, such as skateboarding, roller skating, and riding push scooters, however, have been rather under-researched thus far [12].

Walking (WT) and cycling transportation (CT) appear to be the most frequent types of AT in adolescents from Central European countries [13]. WT is especially beneficial for reasons of lower hygienic demands (no need to change clothes), easier estimation of duration, and better communication options, but also in terms of the possibilities of connecting PA with the use of modern technologies, e.g., listening to music or even educational activities (e.g., preparation for school, language or other types of self-education). Greater autonomy, independence, personal freedom and greater choice are other benefits of WT [14]. Adolescents’ WT to and from school can significantly facilitate the adoption of AT habits in adulthood [9] and, together with the promotion of accompanying positive emotions and an awareness of benefits (economic, relaxation and others), it might increase the odds for acquiring a lifelong positive attitude towards AT [15]. Due to the decline in PA [16,17,18] and AT [1] with adolescents’ age, the adoption of AT habits, awareness of benefits associated with AT and the acquisition of transport literacy are essential both for their current and future healthy lifestyles.

According to the Healthy People 2020 [19], children and adolescents should achieve at least 1 mile when walking or 2 miles when riding a bike. The distance from school is a decisive factor in terms of meeting these recommendations [20,21]. Rodriguez-López et al. [22] suggested 0.84 miles as an acceptable distance for walking to school in adolescents, and according to Chillón et al. [23], acceptable distances are 1.4 km for 10-year-olds, 1.6 km for 11-year-olds and 3 km for 14-year-olds. Similarly, Duncan et al. [24] found the largest AT increase in 5–16-year-old children and adolescents who lived approximately 2 km from their school. It is also very important to note that AT in a ‘friendly’ and safe environment is a very effective method of engaging in PA [25]. In addition, a safe environment for AT increases the odds of meeting the leisure-time walking recommendations, especially in girls [26]; however, the risk of injury increases with distance to school [27].

In children [28] and also adolescents [6,29], it has been shown that AT is associated with a higher all-day level of moderate-to-vigorous PA (MVPA), and that those commuting actively tend to be more physically active than passive commuters [30].

Compared with WT, CT to and from school is less common and less safe in New Zealand [7], but the country differences in WT and CT are substantial [13]. The preference for CT over WT is most pronounced in traditionally ’bicycling’ countries, such as the Netherlands and Denmark [31].

Currently, the emphasis is being put on respecting individual, psychosocial and environmental factors when promoting adolescents’ AT [32]. Despite such efforts, interventions aimed at increasing the overall PA and thus improving the well-being and life satisfaction of adolescents do not always yield the expected effect [33]. Thus, the contribution of AT for meeting the weekly PA recommendations in adolescents is very important [26]. Identifying the mechanisms through which PA increases well-being among adolescents seems to be critical to promote mental health in youth [34].

Markedly different continental, demographic, socioeconomic and educational system settings for adolescents’ AT highlight the importance of considering Central European particulars when promoting AT. Therefore, the aim of this study is to identify the associations between active travel and well-being, the share of active travel in adolescents’ weekly PA, and the contribution of adolescents’ weekly active travel to meeting the weekly PA recommendations in Czech and Polish adolescents.

## 2. Materials and Methods 

### 2.1. Participants and Settings

This retrospective cross-sectional study was carried out between 2014 and 2017 at 36 Czech and 39 Polish schools using the web application ’International Database for Research and Educational Support’ (Indares) (www.indares.com). The methodological background is based on a socioecological model of healthy behavior [35] and the 3P Model: A General Theory of Subjective Well-Being [36]. The schools were selected from a set of schools by quota sampling to reflect the ratio of the main types of secondary schools (grammar, vocational and professional schools). We excluded sports-oriented schools from the selection. Only two schools in Poland refused to participate in research due to the initiation of the new European General Data Protection Regulation. Research in schools took place under the more or less congruent educational and weather conditions of autumn (September–November) and spring (March–May). The respondents completed the questionnaires in information and communication technology classrooms under the guidance of the same research teams in both countries. In total, 1110 boys and 1695 girls aged 15–19 years took part in the research (Table 1). School management, parents and participants provided their written informed consent with research within the school program.

### 2.2. Measurements

#### 2.2.1. Subjective Estimation of Weekly PA

The weekly PA of participants was examined using the Czech and Polish versions of the ‘International Physical Activity Questionnaire—long form’ (IPAQ-LF) [37], which is an expansion of the short IPAQ [38] and allows for more detailed analyses of the weekly structure of PA in young and middle-aged adults (15–69 years). Its use for adolescents aged 15–17 significantly correlated for time spent in active travel, for moderate and vigorous activities as well as for total physical activity [39,40]. Both versions were subject to the required translation procedure according to the ‘EORTC (European Organisation for Research and Treatment of Cancer) Quality of Life Group’ [41] and empirically verified in international comparative studies [26,42]. Among the several types of PA measured, the IPAQ-LF includes transport PA (cycling and walking activities for at least 10 minutes at a time to go from place to place). Since our experience and empiric results indicate the overestimation of time spent doing vigorous PA and the underestimation of time spent sitting [43], and the data adjustments according to the IPAQ-LF manual affect the composition of weekly PA, our procedure was as follows: a) compared to the IPAQ-LF manual, we multiplied the MET-min of vigorous PA (VPA) by six instead of the recommended multiplication by eight to reduce overestimation in this variable; b) estimated minutes in each type of PA, sitting and commuting (travel in a motor vehicle, e.g., train, bus, car, or tram) over a week were converted to average minutes of PA per day; c) we set the permissible average daily sum of PA and transportation at 600 minutes; The maximum MET-min per week was set at 20,000 MET-min; d) the maximum average daily sum of PA, transportation, sitting and passive commuting was set at 960 minutes. We removed 657 respondents from the dataset for non-compliance with these criteria.

We modified the recommendations for weekly AT according to Healthy People 2020 [19] and the Physical Activity Guidelines for Americans [44]. We selected the minimum recommendation because it was only based on a single given type of PA in the IPAQ-LF questionnaire. The minimum recommendation was at least five or more days per week for 30 or more minutes (five times 30 minutes) for CT, WT, as well as for aggregated AT (either CT or WT). The recommendations for weekly PA of at least five or more days a week for 60 minutes of MVPA and at the same time, on three or more days a week for at least 20 minutes of VPA (five times 60 min PA and three times 20 min VPA). Due to its difficulty, the IPAQ-LF questionnaire was the first to be completed. Researchers’ presence during the filling in the questionnaire was necessary, since the structure of the IPAQ questionnaire, as well as its contents, are complex and might require further explanation.

#### 2.2.2. Subjective Estimation of Sedentary Behaviours (Passive Commuting and Sitting)

The time spent in passive commuting and sitting was also observed from IPAQ-LF. Time spent in passive commuting was adjusted to be in agreement with the IPAQ-LF manual. The average daily sum of sitting was limited to a maximum of 600 min/day. The adjustment of sedentary time was done for 286 participants. The time of sedentary behavior in minutes was set as the summation of time spent in passive commuting and sitting. Two groups, one of less (<Mdn in terms of time spent commuting and sitting) and one of more (>Mdn in terms of time spent in passive commuting and sitting) sedentary participants were created.

#### 2.2.3. Self-Reported Well-Being

The adolescents’ well-being was assessed using the WHO-5 Well-Being Index (1998 version) in the Czech and Polish versions (https://www.psykiatri-regionh.dk/who-5/Pages/default.aspx). It contains five questions with a total gross score of 25 points. A score of less than 13 points represents a lower WB and a score of ≥13 indicates a higher WB.

Compliance with recommendations for weekly PA of at least five or more days a week for 60 minutes of MVPA and, at the same time, on three or more days a week for at least 20 minutes of VPA (five times 60 min PA and three times 20 min VPA) altogether with a well-being index score of ≥13 points represented a healthy and physically active lifestyle of adolescents.

### 2.3. Data Analysis

The Statistica version 13 (StatSoft, Prague, Czech Republic) and SPSS version 25 (IBM, Armonk, NY: IBM Corp.) software were used for the statistical analyses. We used descriptive characteristics and cross tables to assess the differences in compliance with the PA recommendations. The Kruskal–Wallis ANOVA was used to investigate differences in PA by WB level. To retrieve the odds of meeting the PA recommendations, we applied binary logistic regression analyses with the enter method (all independent variables are entered into the equation at the same time). In all models, the first indicators in categorical covariates were used as references. The practical significance was estimated using the η^2^ and w coefficients, which were interpreted as follows: η^2^: 0.01–0.059 small, 0.06–0.139 medium and ≥0.14 large effect size; w: 0.1–0.29 small, 0.3–0.49 medium and ≥0.5 large effect size.

### 2.4. Ethical Statement

The study was approved by the Ethical Committee of the Faculty of Physical Culture of Palacký University Olomouc under No. 37/2013. School management faculties, parents and participants confirmed their agreement to participate in the research by written consent. They were informed about the security and anonymity of the data obtained by the Indares web application, the method by which it would be processed and its further use. Across the participating schools, management faculties and participants were notified of the average group results regarding the rates of meeting the AT and weekly PA recommendations.

## 3. Results

### 3.1. The Associations Between Active Travel and Well-Being

We found a statistically significant association of CT with lower (M = 181 METs-min/week) and higher (M = 272 METs-min/week) WB (*p* = 0.033) in girls from both countries (Table 2). The statistically significant association of overall weekly PA with lower WB (5122 METs-min/week) and higher WB (5808 METs-min) was observed in boys’ overall PA aggregated by both countries (*p* = 0.047) and, similarly, in girls with lower WB (4388 METs-min/week) and higher WB (5094 METs-min/week) (*p* ˂ 0.001).

The overall AT of Czech girls (1228 METs-min) and boys (1209 METs-min) represents 24.9% and 22.5% of their total weekly PA, respectively. Similarly, AT accounts for 24.5% of total weekly PA in Polish girls (1262 METs-min) and 24.2% of total weekly PA in Polish boys (1409 METs-min).

### 3.2. The Association Between Active Travel Recommendation and Well-Being

Polish girls who meet the active travel recommendation (ATR) and report higher WB are statistically significantly more likely to meet the overall weekly PA recommendation (40% of girls), compared with 29% of girls not meeting the ATR and indicating a lower level of WB (χ^2^ = 4.52, *p* = 0.034, *w* = 0.075) (Figure 1).

Similarly, significantly more Polish boys who meet the ATR and report higher WB also met the overall weekly PA recommendation (54% of boys), as opposed to 26% of boys not meeting the ATR and having lower levels of WB (χ^2^ = 16.45, p ˂ 0.001, *w* = 0.169 ^*^) (Figure 1). We found no statistically significant differences between meeting the ATR in Czech girls with lower (21% girls) and higher (28% girls) levels of WB (*p* = 0.100), or in Czech boys with lower (33% boys) and higher (35% boys) levels of WB (*p* = 0.725).

### 3.3. Odds of Meeting the Weekly PA Recommendations According to Meeting the Active Travel Recommendations and Level of Well-Being

Girls and boys who met the ATR and concurrently reported higher WB were more likely to meet the weekly PA recommendations (five times 60 min or PA and three times 20 min of VPA) than those who did not meet the ATR and had lower WB (boys’ OR = 2.729, CI = 2.077–3.584, *p* < 0.001; girls’ OR = 2.448, CI = 1.920–3.121, *p* <0.001). Control variables in Model 2 (BMI and age) and Model 3 (country, size of city, type of housing, dog in a family, car in a family, and participation in organized PA) did not affect the significance of these odds. The influence of particular variables on associations between meeting the weekly PA recommendations and meeting the ATR and higher WB is presented separately for girls and boys in Figure 2. Among control variables, participation in organized PA had the most significant influence on meeting PA recommendations. In all, 67.0% of respondents participating in organized PA also showed higher WB, while the total was only 57.3% for those not involved in organized PA (χ^2^ = 23.32, *p* ˂ 0.001, w = 0.091).

### 3.4. The Associations Between Sedentary Behaviour (Commuting and Sitting) and Well-Being

In both countries, girls and boys who were more sedentary reported having lower WB than those who spend less time being sedentary (Table 3). However, the differences were only statistically significant in Czech girls (75.6% vs. 51.5%) and Polish boys (66.4% versus 57.1%) (Table 3).

## 4. Discussion

### 4.1. The Active Travel Recommendation and Well-Being in the Structure of the Weekly PA

Our finding that AT accounts for 24.9% of overall weekly PA in Czech girls (Polish 24.5%) and for 22.5% in Czech boys (Polish 24.2%) confirms the generally recognized potential of AT in contributing to and supporting overall PA [45,46]. It is vital to maintain this share of AT in overall PA, especially due to the increase in private car transport usage, a trend which Central and Eastern European countries have been following in relation to more developed countries. Both girls and boys who met the ATR and reported higher WB at the same time are more likely to meet the weekly PA recommendations (five times 60 min of PA and three times 20 min of VPA) than those who do not meet the ATR and have lower WB. This is particularly important considering that, although WB and PA decrease during adolescence [47], especially in girls [48], meeting ATR might help to increase the level of PA and WB as well. It was further confirmed that meeting the ATR in combination with having a higher WB does not have to occur at the expense of meeting the VPA recommendations. This is also essential, because significant associations were observed between VPA, mental well-being and overall quality of life in adolescents [49].

Given that trends in adolescents’ participation in organized PA are inconsistent [1] and tendencies of participation in organized PA with age are not fully clear either, it is crucial that the compliance with ATR does not act as a substitute for compliance with VPA recommendations. Our finding that higher WB is associated with the participation of adolescents in organized PA is serious and corresponds with other research [50]. It is apparent that the factor of organized PA, in combination with higher WB, plays a more important role in meeting the PA recommendations in Central European settings than other sociodemographic factors we monitored—e.g., country, size of the city, type of house, and the ownership of a dog or car. For this reason, the emphasis on promoting adolescent involvement in school sports is also very important, especially for girls [51].

### 4.2. The Association Between Active Travel and Well-Being

Even though there is still only limited evidence of the contribution of AT to adolescent health [52], several studies confirm positive associations between AT and well-being [53,54]. Adolescents who are not physically active have lower levels of life satisfaction than those who are physically active [47]. Our finding that associations between AT and WB are similar in girls and boys does not diminish the importance of respecting gender specifics in these associations [55]. In our sample, there was a greater proportion of boys indicating higher WB (70.7%), compared with girls (60.0%). However, these are the differences in a simplified short-term assessment that disallows the generalization of this comparison. Therefore, Gill et al. [56] recommend examining the relationships of PA with both an integrated and dimensional (social, emotional, cognitive, physical, spiritual and functional) approach to WB. Nonetheless, the benefits of different types of transportation PA for dimensionally structured WB have scarcely been researched thus far. Furthermore, the basic theoretical background of the General Theory of Subjective Well-Being in the form of the 3P Model [36], which is based on time components (Past, Present, and Prospect), should be respected. The continuity of the preferred type of AT and the safety of the environment for the preferred AT should be conducive to particular physical, social, emotional, but also environmental WB, thereby increasing the chances of sustaining and amplifying adolescent WB. This is likely to be supported in adolescent age, especially in physically and socially friendly, timesaving and safe environments [26]. The current young people’s initiatives to combat climate change are increasing the likelihood of them supporting these positive changes as well. It is also necessary to promote AT in a natural or nature-like environment as much as possible. This is because outdoor PA contributes to WB more than PA in a ‘non-natural’ environment [57,58]. The development of natural areas is gradually affecting the environment around schools and often presenting an obstacle to the support for children’s and adolescents’ AT. Unfortunately, the current interventions aimed at increasing PA and the promotion of WB do not always deliver the intended effect [33]. The effects of interventions to increase active school transport vary and the quality of evidence remains low [59]. Well-separated and safe cycling infrastructure seems to be more important to adolescents than distance and the social environment [60]. We can also expect more health benefits in adolescents who cycle than in non-cyclists [9]. For example, the fact that 66% and 71% of Danish adolescents actively commute to and from school, respectively, and that WT and CT account for 64% of AT [61] should be encouraging for other countries with similar demographic settings, including those from Central Europe. Given the number of determinants that affect the choice of WT or CT, these two types of AT should be perceived as distinguished alternatives [5].

### 4.3. The Associations Between Sedentary Behaviour and Well-Being

In our sample, both girls and boys who were more sedentary reported lower WB than those who spent less time being sedentary, however these associations were not statistically significant except with regard to Czech girls and Polish boys. Although we cannot generalize results, similar findings were observed in a Spanish [55] and Swedish study [62] where a lower level of PA was associated with lower WB in both girls and boys. A Norwegian study [63] identified the lack of general well-being and lack of sports and exercise as potential predictors in self-rated health, especially in girls, who reported subjective health deterioration more often than boys. Through simple questions on sedentary behaviour, we wanted to find out basic information about these associations in adolescents from the Central European region. Deeper analyses to obtain valid data is needed in future research.

### 4.4. Active Travel in the Context of the Policy

The unclear effects of AT interventions [64] are particularly challenging when determining state, municipal and school policies. Furthermore, they represent a challenge for school management faculties, teachers and parents to enact positive changes that consider the various specifics of the built environment, transport, region, socioeconomic differences especially in developing curricular and extracurricular programs at different types of schools. Equally important is the active involvement of adolescents in creating the proposals to promote changes in AT and increase the sense of co-responsibility and pride for the positive changes achieved. Stimuli to promote and increase adolescents’ AT, not limited to transportation to and from schools, should be integrated into the Comprehensive School Physical Activity Programs [65]. It is highly important to accelerate the transfer of knowledge to national, regional and school practice, in spite of there being insufficient evidence-based knowledge regarding transportation [66]. Given the similar sociodemographics, health and educational development of other Central and Eastern European countries, it is very likely that results of this study will also be beneficial for these countries. The practical reality is that negligence toward the AT environment is very difficult to address politically and is economically expensive. However, building safe routes to schools has not only a health and psychological impact, but also an economic effect, especially when a large portion of the student body live close to the school [67].

Further research should focus on the combination of objective monitoring and qualitative assessment of different types of AT, in the context of the safety as well as on sedentary behavior as a possible predictor for low well-being in adolescents.

### 4.5. Strengths and Limitations

The strength of the study lies in the novelty of assessing the associations between AT recommendations, weekly PA recommendations, and the different levels of well-being in two Central European countries. Furthermore, another strength is the presentation of broader possibilities of using the results from the IPAQ-LF questionnaire.

The limitations of the study include the sampling of secondary school students, as they were selected (deliberately) even though in an identical way in both countries. In addition, for the weekly PA and sedentary behavior estimate, allowing an assessment of students’ compliance with the AT and weekly PA recommendations using only a single type of PA is limiting, because the IPAQ-LF does not enable us to distinguish individual days of the week. The direction of influence could not be determined because of the cross-sectional design of the study.

## 5. Conclusions

Active travel, in conjunction with adolescents’ well-being, increases the odds of meeting the physical activity recommendations in secondary-school-level adolescents. The active travel levels of boys and girls in both countries account for 22.5% to 24.9% of their overall weekly PA. Meeting the active travel recommendations should be part of comprehensive school-based PA programmes. State, school and municipal policies in Central European countries should focus on maintaining and improving conditions for the adolescents’ preferred type of active travel in an emotionally favourable and safe environment.

## Figures and Tables

**Figure 1 ijerph-17-02001-f001:**
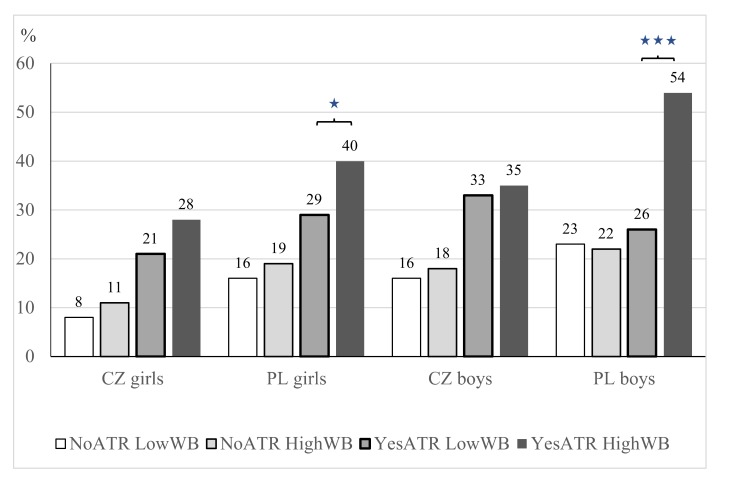
Rates of meeting the weekly physical activity (PA) recommendations (at least five times 60 min of moderate-to-vigorous PA (MVPA) and at least three times 20 min of vigorous PA (VPA)) in Czech (CZ) and Polish (PL) girls and boys by level of well-being (WB) and meeting active travel recommendation (ATR) (at least five or more days per week for 30 or more minutes of active travel (AT)).

**Figure 2 ijerph-17-02001-f002:**
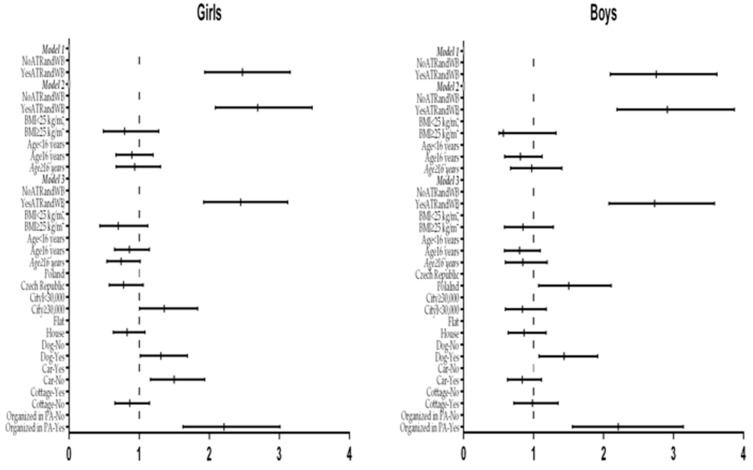
Odds ratios for meeting the weekly PA recommendation (five times 60 min of MVPA and concurrently three times 20 min of vigorous PA) in girls and boys according to compliance with the meeting of active travel recommendation along with greater well-being (YesATRandWB) or non-compliance with meeting the ATR and lower well-being (NoATRandWB). *Notes:* Model 1 = yes—meeting the active travel recommendation and high well-being; no—not meeting the active travel recommendation and low well-being; Model 2 = adjusted for age and BMI; Model 3 = adjusted further for country, city, house, ownership of dog, car, and cottage, and organized PA.

**Table 1 ijerph-17-02001-t001:** Sample characteristics.

Characteristics	*n*	Age (Years)		Weight (kg)		Height (cm)		BMI (kg·m^-2^)	
		*M*	*SD*	*M*	*SD*	*M*	*SD*	*M*	*SD*
**Girls CZ**	931	16.77	1.18	59.49	9.26	167.50	6.42	21.18	2.95
Boys CZ	531	16.74	1.22	70.35	11.85	179.06	7.86	21.92	3.37
Girls PL	764	16.26	0.77	56.95	8.79	166.10	6.07	20.62	2.85
Boys PL	579	16.21	0.74	67.02	13.37	176.63	7.74	21.41	3.65

*Note:* mean (*M*); standard deviation (*SD*); body mass index (BMI); Poland (PL); Czech Republic (CZ).

**Table 2 ijerph-17-02001-t002:** Weekly transportation and overall physical activity in girls and boys by their well-being (WB) level.

Physical Activity	Girls				Boys				H	*p*	η^2^
	Low WB (*n* = 678)		High WB (*n* = 1017)		Low WB (*n* = 325)		High WB (*n* = 785)				
	Mdn (IQR)	M (SD)	Mdn (IQR))	M (SD)	Mdn (IQR)	M (SD)	Mdn (IQR))	M (SD)			
Cycling transportation (MET-min/week)	**0** **(0)**	**181** **(703)**	**0** **(60)**	**272** **(827)**	0 (120)	308 (763)	0 (360)	462 (1104)	66.34 ^b,c,d^	˂0.001	0.037^*^
Walking transportation (MET-min/week)	57 8(1122)	910 (989)	578 (1139)	981 (1074)	462 (990)	874 (1061)	495 (990)	906 (1058)	10.07	0.018	0.004
Total transportation (MET-min/week)	693 (1116)	1090 (1217)	743 (1424)	1253 (1333)	675 (1544)	1181 (1312)	792 (1679)	1368 (1587)	7.90	0.048	0.003
Total physical activity (MET-min/week)	**3485** **(4274)**	**4388** **(3354)**	**4102** **(5103)**	**5094** **(3645)**	**4212** **(5362)**	**5122** **(3787)**	**5014** **(4051)**	**5808** **(4051)**	47.76 ^a,b,c,d^	˂0.001	0.037^*^

*Note**:* number (*n*), median (Mdn), interquartile range (IQR), arithmetic mean (M), standard deviation (SD), Kruskal–Wallis test (H), level of significance (*p*), effect size coefficient (η^2^), small effect size (/^*^), significant differences between boys with low WB and high WB (^a^/), significant differences between girls with low WB and high WB (^b^/), significant differences between boys with low WB and girls with low WB (^c^/), significant differences between boys with high WB and girls with high WB (^d^/).

**Table 3 ijerph-17-02001-t003:** The associations between sedentary behaviours (passive commuting and sitting) and well-being.

Country	Gender	Well-being	Passive Commuting and Sitting				χ ^2^	*p*	*w*
			Low		High				
			*n*	*%*	*n*	*%*			
Czech Republic	Girls:	Low	117	24.4	48	48.5	23.41	<0.001	0.201 ^*^
		High	363	75.6	51	51.5			
	Boys	Low	124	29.3	36	34.6	0.79	0.375	0.039
		High	300	71.7	71	66.4			
Poland	Girls:	Low	258	43.8	85	48.6	1.24	0.265	0.041
		High	331	56.2	90	51.4			
	Boys	Low	233	33.6	102	42.9	6.56	0.010	0.084
		High	460	**66.4**	136	**57.1**			

*Note:* Pearson’s chi-squared test (χ^2^), statistical significance (*p*), Cohen’s effect size coefficient (*w*), small effect size (/^*^).

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
