# Peer review of "Active Travel of Czech and Polish Adolescents in Relation to Their Well-Being: Support for Physical Activity and Health"

_ijerph, 2020, doi:10.3390/ijerph17062001_

Round 1
Reviewer 1 Report
This is an interesting paper based on analysis of a solid dataset of questionnaire responses on active transport, physical activity and wellbeing by 2,805 adolescents in the Czech Republic and Poland. The paper is written with reference to many recent papers on the topic. The findings vary for different groups of respondents.
The authors need to decide whether the main question being addressed is: 1. the association between walking and cycling with wellbeing, as in the current title of the paper, or 2. the association between active travel and wellbeing with physical activity, as in the Conclusions. Depending on the answer, the title and/or some of the text will need to be refined.
The authors make some comments from the literature on the physical qualities of environments that encourage active transport, whether it is natural, safe, well separated, physically and socially friendly (lines 261 -284). Those attributes might be included in the authors' suggestions for state, municipal and school policies related to encouraging AT for adolescents.
The authors might also wish to comment on:
- whether they feel that any of the findings might be generalisable to other countries
- further research that needs to be done eg validation of the active travel and physical activity questions by use of acceleromoters
- possibly adding a note somewhere in the text on the increasing understanding of 1. the link between forms of travel, local air quality/pollution close to schools with consequent effects on health, and 2. longer term planetary health related to climate change hence potentially adding an additional reason for the value of active travel to the wider society, not just to the individual
Below are noted some minor suggestions for improving the clarity of the language and presentation of the paper:
- The title of the paper should also include a reference to physical activity, also that forms an important part of the analysis.
- "lower hygienic demands" (line 42) express more clearly
- "that" (line 58) should be "who"
- "active commuters . . . inactive commuters" - unclear needs to be rephrased (lines 63-64)
- Section 2.1: Add a note on how many schools that were contacted refused to participate, and on how many pupils in the participating schools refused to participate.
- Lines 103-104 Clarify whose experience and empirical results are being discussed, as ref. 39 does not include any of the authors of this paper.
- Line 107 Explain why a factor of 6 was selected
- Line 121 Is 'observed' the right term?
- Lines 161-165. Could be expressed more clearly
- Combine Figs 1 and 2 into one chart
Fig 3 Title and explanation could be more clearly expressed
Line 261 Alter "safeness" to "safety"
Consider including a single box or footnote providing an index of all abbreviations used in the paper.
Reviewer 2 Report
1
Your paper shows an clear set up to compare the PA and WB of 2 teenage groups by means of self reporting which are corrected because people may use under and overestimation. i like that . it is like asking people what their height and weight is. people will lie towards the groups mean value. of course better is measuring it by an physical instrument.
2
also your paper takes care of spring an autumn influences. i like that too. but i would expect also a paragraph about the difference in results between spring and autumn or perhaps it does not matter at all. even then i would like to see this reported.
3
figure 3 is unclear for the reader; the distance between the left side column and the 2nd column with results, is too far to draw conclusion. for example if i want to read the difference for row BMI<25 and row BMI>25 I canot see which results belongs to which row. please redesign this figure.
Reviewer 3 Report
Walking and Cycling Transportation of Czech and 2 Polish Adolescents in Relation to Their Well-Being.
This work is an interesting study that associates active transport with well-being, with an interesting qualitative perspective.
Here are some comments that can help improve the document.
Key words
Line 27-28: try to write words and not sentences like keywords.
Introduction
Line 76: use PA to refer to physical activity.
Line 77: there is talk about meeting the weekly PA recommendations, but before it is not explained what these recommendations are.
Line 78: Apparently “adolescents” It is written with a larger size.
Materials and methods
Line 88: is necessary to explain “ICT”
Line 92: Table 1. Order per country and put first “girls” and after “boys”.
Line: 98: Here is great methodologic problem. The IPAQ was designed to be used by adults aged 18–65 yr. In references 35 and 36, this stands out.
The authors must justify very well the use of this instrument designed for adults, which has been used in adolescents.
Otherwise there is a high risk that this article cannot be published for this reason.
Line 107: was changed only VPA? The Moderate PA was multiplicated by 4 or was modified? To explain better.
Line 121: the word commuting is used for referring toward all type of commuting. I suggest that this be clarified. It could be added for example: "passive commuting" or "motorized commuting" etc.
Line 134-135: This information does not match what is stated in lines 115-116. I suggest that this be clarified.
Results
Line 161-165: Does the METs / min correspond to the week? (METs / min / week?)
Line 167: organize table 2 and put girls first.
The table 2 is difficult to read. The presentation and organization of data should be improved, as well as the presentation of statistical differences.
Line 180: the meet active transportation recommendation (ATR) was not explained in methods.
Line 185-194: At the bottom of the figure it is noted that "meeting the weekly PA recommendations", but only ATR is presented. Improve the explanation.
Line 214: Figure 3 has a lot information and is difficult to understand.
I suggest that the following changes be made:
- The reference line of 1, be continuous, not dotted.
- On the vertical axis remove the "no" variables. Because it's very saturated. It can be explained separately what variables were considered.
- It is not necessary to highlight (with a thicker line) those variables that were significant, since to see that they are on the reference "1", it is well understood.
Line 230: correct and clarify the meaning of "commuting" that appears in table 3.
I think that no was clarified as was classified low and high “commuting and sitting”.
Discussion
General comments: I suggest that you order the ideas of the discussion, because one and the other ideas are mixed in each paragraph that is read.
To do this you can create subtitles and organize the topics that way.
Line 238: I don't see the coherence of this sentence: "This is particularly important considering that WB, as well as PA, decrease during adolescence", with the results of the study.
Line 241: It is the first time that the term “quality of life” appears. It should be explained better.
Line 259: the general theory of subjective well-being should have been explained in the introduction.
Strengths and limitations
The mean limitation is the instrument used (IPAQ for adults). It must be added.
Conclusions
There is talk separately of “walking and cycling”, but these results are not analyzed independently, but are done together (active transport).
There are two options, the phrases "walking and cycling" are removed, even in the title or make a new analysis separately.
Round 2
Reviewer 3 Report
There are three important points that have not been resolved.
1. I have asked that they reorder the results (Tables, figures), where the girls must join before the boys. This is very important, from a gender perspective.
2. I have asked that the use of IPAQ in adolescents be justified, but the exposed citations range from 18 to 69 years. The authors have justified its use in two American documents, where it has been difficult for me to find. It is preferable that its use be justified through scientific articles.
3. I have asked for a change in the order of the discussion and to improve the structure, which has not been done either.
I do not understand, so that the reviewer can make suggestions for changes and improvements, if later the authors do not assume them.
In my opinion, the article cannot be published in the current version.
